# Performance evaluation of nano-graphene lubricating oil with high dispersion and low viscosity used in diesel engines

Xin Kuang[1], Xiping Yang[2]*, Hua Bian[2‡], Rong Kuang[3‡], Nanrong Hu[3‡], Shengyong Li[1‡]

1 School of Traffic Engineering, Jiangsu Shipping College, Nantong, China, 2 School of Intelligent Manufacturing and Information, Jiangsu Shipping College, Nantong, China, 3 School of Automotive and Traffic Engineering, Jiangsu University, Zhenjiang, China

☯ These authors contributed equally to this work.
‡ HB, RK, NH and SL also contributed equally to this work.
* yxp@jssc.edu.cn

**Data Availability Statement:** All relevant data are within the manuscript.

## Abstract

The basic tribological experiments have reported that nano-graphene lubricating oil has excellent anti-friction and anti-wear properties, which has been widely concerned. However, the real anti-friction effect of nano-graphene lubricating oil and its impact on engine power performance, economic performance and emission performance remain to be proved. This has seriously hindered the popularization and application of nano-graphene lubricating oil in the engine field. In this paper, nano-graphene powder was chemically grafted to prepare nano-graphene lubricating oil with high dispersion stability. The influence of nano-graphene on physicochemical properties of lubricating oil was studied, and the influence of nano-graphene on engine power performance, economic performance and emission performance was explored. The results show that after modification, the dispersion of nano-graphene in lubricating oil is improved. Compared with pure lubricating oil, the addition of nano-graphene makes the kinematic viscosity of lubricating oil slightly lower, and has little effect on the density, flash point, pour point and total acid value of lubricating oil. The reversed towing torque of nano-graphene lubricating oil is reduced by 1.82–5.53%, indicating that the friction loss decreases. The specific fuel consumption of the engine is reduced, which indicates that the fuel economic performance is improved. Engine HC+NO$_X$, CH$_4$, CO$_2$ emissions do not change much, but particulate matter (PM) emissions increase by 8.85%. The quantity concentration of nuclear particles, accumulated particles and total particles of nano-graphene lubricating oil are significantly higher than that of pure lubricating oil. And the increase of the quantity concentration of accumulated particles is more obvious than that of nuclear particles, and the larger the load, the more obvious this phenomenon. In order to apply nano-graphene lubricating oil to the engine, it is also necessary to further study its impact on the post-processing system, adjust the control strategy of the post-processing system and then test and calibrate.

**Funding:** This work was supported by Natural Science Research of Jiangsu Higher Education Institutions of China [grant number 23KJB470006; 23KJB460009]; Science and Technology Project of Nantong City [grant number JC22022066]; Natural Science Foundation of Jiangsu Province [grant number BK20231227]; High-level Talents Research Start-up Fund supported by Jiangsu Shipping College [grant number HYRC/202407; HYRC/202411]; Qinglan Project of Jiangsu Province of China.

**Competing interests:** The authors have declared that no competing interests exist.

## 1. Introduction

Industry experts believe that the engine will remain viable for a long time and will remain the most important power source for passenger cars in 30 years [1]. Lubricating oil is known as the "blood" of the engine and is one of the key factors affecting the mechanical efficiency, reliability and emissions of PM of the engine [2–4]. In recent years, the urgent needs of near zero emissions and low fuel consumption of engines have put forward higher requirements on the quality of lubricating oil, such as low ash content, low sulfur and phosphorus, low viscosity, long oil change cycle, anti-friction and anti-wear performance, etc. [5–8], especially excellent anti-friction and anti-wear properties. With the development of nanotechnology, nano-additives have been applied to the field of industrial lubrication, such as soft metals, metal compounds, organic compounds, graphite and its interlayer compounds [9–13]. Among them, nano-graphene, the thinnest two-dimensional nano-carbon material that only contains carbon elements, has attracted attention as a lubricating oil additive [14–17]. Nano-graphene has a unique nanolayered structure, forming a transfer film at the friction interface. Good tribological properties are obtained due to the self-lubrication of graphene by sliding between layers [17].

A large number of basic tribological experiments have found that the addition of an appropriate amount of nano-graphene as a general industrial lubricating oil additive can play a role in reducing friction and wear [18–23]. In recent years, researchers have begun to study the addition of nano-graphene to engine lubricating oil, mainly focusing on its dispersion stability and tribological properties.

Alqahtani et al. selected dimethylformamide as solvent, mixed nano-graphene into SAE 5W-30 lubricating oil, and studied the tribological effects of graphene nanosheets on engine SAE 5W-30 lubricating oil by using a four-ball friction tester. The results showed that the wear marks and friction coefficients were improved by 15% and 33% respectively after 0.12 wt.% nano-additive was added [15]. Nowduru et al. investigated the effects of nano-graphene on the tribological properties of engine oil 15W-40 using a four-ball test in accordance with ASTM D4172. The results showed that the mean coefficient of friction at room temperature and 75˚C was reduced by 25.5% and 16%, respectively. And the wear resistance at room temperature and 75˚C was increased by 13% and 16%, respectively [24]. Using a reciprocating tribometer to simulate the ring-liner interfaces of an engine, Ali et al. investigated the effects of graphene nanoparticles on the tribological properties of a fully formulated engine 5W-30 lubricating oil at different speeds and loads in accordance with ASTM G181-11. The experimental results showed that compared with 5W-30 lubricating oil, nano-additive can reduce the wear rate and friction coefficient by 25% ~ 30% and 26.5% ~ 32.6%, respectively [25]. Franzosi et al. investigated the tribological properties of functionalized graphene nanoplatelets added to fully formulated engine lubricating oil using a continuous sliding/rolling ball-on-disk tribometer. Friction tests showed that 0.2 wt% graphene lubricating oil reduced friction by 28% and 11% in the boundary and elastohydrodynamic lubrication regimes, respectively [26]. In the previous stage, Kuang et al. of our research group used oleic acid and stearic acid to chemically modify nanographene, and studied the tribological properties of the engine's key friction pair, the cylinder liner and piston sample, under the action of lubricating oil on a reciprocating high-frequency friction test machine. The results showed that adding 25 ppm of nano-graphene to lubricating oil had the best frictional and wear reduction effect. Compared with pure lubricating oil, at 25˚C and 100˚C, the friction coefficient was reduced by 6.37% and 17.72%, and the wear amount was reduced by 8.18% and 25.19%, respectively.

Nano-graphene has broad application prospects. However, as an engine lubricating oil, different from industrial lubricating oil for general use, it is currently required to transition to

Euro VI, national VI and above ultra-low emission regulations. The complicated and extremely harsh working conditions of the engine have brought higher challenges to the dispersion stability, physicochemical properties of nano-graphene lubricating oil and the anti-friction and anti-wear effect under real working conditions. At the same time, the impact of nano-graphene lubricating oil consumption on emissions and post-treatment system reliability deserves attention. There is little research on these areas. Ali et al. added graphene with a sheet diameter of 5–10 μm and a thickness of 3–10 nm to 5W-30 lubricating oil at a mass concentration of 0.4%. European driving cycle tests on graphene lubricating oil and pure lubricating oil was carried out respectively, and the characteristics of engine emissions at different speeds and loads under the same conditions were compared. The results showed that: under most working conditions, the emission performance of graphene lubricating oil was improved, $CO_2$ emission was reduced by 3.4–4.66%, $NO_X$ was reduced by about 3–5%, and HC was reduced with the increase of engine load [27]. This shows that the addition of nano-graphene will have a certain impact on $CO_2$, HC, $NO_X$ emissions, but there is a lack of attention to the dispersion of nano-graphene in lubricating oil, the physicochemical properties of nano-graphene lubricating oil, engine friction loss and PM emissions.

Therefore, from the perspective of engine application, the nano-graphene additive with the most potential and representative to reduce friction and wear was selected to be introduced into the engine low-viscosity lubricating oil. Based on engine performance test platform and multiple characterization methods, the dispersion stability, physicochemical properties, and real anti-friction effect of nano-graphene lubricating oil were studied. And the potential effects of nano-graphene additives on engine power performance, economic performance, emission performance, PM particle size distribution and post-treatment system were investigated, which laid a theoretical and scientific basis for the design, development and application of nano-graphene lubricating oil.

## 2 Experimental section

### 2.1 Experimental prototype and materials

The diesel engine model used in the experiment is KD192FW, and the relevant technical parameters are shown in Table 1. Commercially available 0# National 5 diesel was selected for the experiment, and its physicochemical properties are shown in Table 2. Nano-graphene was commercially available, and their transmission electron microscopy (TEM) photographs are shown in Fig 1. Through the measurement of Digital Micrograph software, it can be seen that the average thickness of nano-graphene powder is about 0.5–1 nm, and the number of nano-graphene layers is about 1–3 layers. The nano-graphene was chemically modified by oleic acid and stearic acid [28] as follows. First, 0.5 g of nano-graphene was dispersed into 100 mL of anhydrous ethanol. 2 g stearic acid and 3 g oleic acid were then added into them. Finally, the oil-soluble nano-graphene was obtained by centrifuge drying after stirring at 80˚C for 4 hours.

**Table 1. KD192FW diesel engine related technical parameters.**

| Parameter | Index |
| --- | --- |
| Form | Single cylinder, four stroke, air cooled, vertical |
| Cylinder diameter | 92 mm |
| Piston stroke | 75 mm |
| Rated power/speed | 7.6 kW/3000 r·min$^{-1}$ |
| Maximum torque/speed | 25 N·m/2500 r·min$^{-1}$ |
| Compression ratio | 19.5 |

**Table 2. Main physicochemical properties of diesel for experiment.**

| Parameter | Index |
|---|---|
| Density (20°C, kg/m$^3$) | 0.836 |
| Flash point (°C) | 57 |
| Low calorific value (MJ/kg) | 42.90 |
| Ash content (%) | 0.001 |
| Cetane number | 51.5 |
| Kinematic viscosity (20°C, mm$^2$/s) | 5.055 |

Previous experiment results [28] showed that the modified nano-graphene lubricating oil with a mass concentration of 25 ppm has the best tribological properties. The modified nano-graphene with a mass concentration of 25 ppm was added to the lubricating oil by weighing it in a precision balance. Then it was sufficiently intermittently dispersed at a low temperature after strong stirring by a magnetic mixer and a high-frequency ultrasonic disperser until it was steadily dispersed in the lubricating oil to obtain nano-graphene lubricating oil. It was recorded as MGL25. The pure lubricating oil used in the experiment is low-viscosity SN/CF 5W-30 diesel engine lubricating oil, labeled PLO, and its physicochemical properties parameters are shown in Table 3. The lubricating oils used in the experiment were PLO and MGL25 respectively.

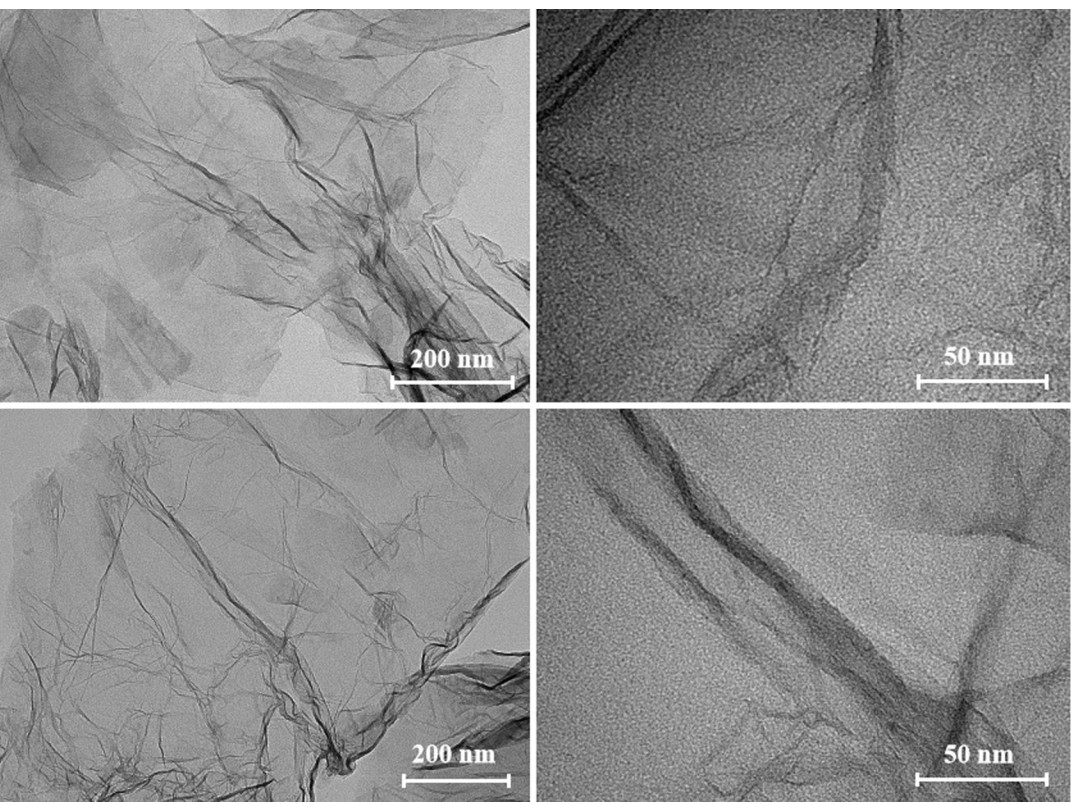

**Fig 1. TEM photographs of nano-graphene.**

**Table 3. Physicochemical property parameters of PLO.**

| Density (20°C) (kg/m³) | Flash point (opening) (°C) | Pour point (°C) | Total acid number (in KOH) (mg/g) |
|---|---|---|---|
| 0.893 | 225 | -48 | 1.22 |

## 2.2 Experiment of physicochemical properties of lubricating oil

The dispersion of freshly prepared lubricating oil with modified nano-graphene and unmodified nano-graphene after 1 week of placement were observed, respectively. Their absorbance was recorded periodically with the UV2600-204 UV-VIS spectrophotometer. Each measurement was repeated at least 3 times and the average was taken to get the final result. The data was converted to the corresponding relative concentration to quantitatively determine its dispersion stability. The upper layer of nano-graphene lubricating oil was taken for physicochemical properties measurement, such as density, flash point, pour point, total acid value, viscosity-temperature characteristics. The relevant measurement standards and instruments are shown in Table 4. The thermogravimetric (TG) curves of different lubricating oil samples were measured by TGA/DSC1 thermogravimetric analyzer. PLO and MGL25 samples were respectively weighed 2 mg. 80% $N_2$ and 20% $O_2$ was selected as the test reaction gas, and high-purity $N_2$ was used as the protection gas. The gas flow rate was 50 mL/min. The temperature was increased from room temperature to 400°C, and the heating rate was set at 10°C/min. The oxidation characteristics of lubricating oil were analyzed according to the TG curves.

## 2.3 Engine reversed towing experiment

**2.3.1. Experiment device and instrument.** The schematic diagram of engine reversed towing experiment device is shown in Fig 2. The experiment bench system mainly includes experiment prototype, electric eddy current dynamometer, dynamometer control system, lubricating oil and exhaust temperature meter. The main experiment device and instrument are shown in Table 5. The start and stop of diesel engine and the regulation of operating conditions were completed by the control system EST2010.

**2.3.2. Experiment scheme.** Reversed towing experiment was carried out on the experimental prototype, and the reversed towing torque at different speeds was measured to compare the friction power consumption lubricated with two different lubricating oils. During the experiment, the throttle valve was kept fully open to reduce the error caused by the pump air loss, so that the friction between the piston assembly and the cylinder liner accounts for the main part of the reversed towing torque. The specific experiment process is as follows. After idling the engine for half an hour, when the lubricating oil temperature reached 80°C, the fuel supply to the engine was cut off and the reversed towing experiment was carried out. During the reversed towing experiment, the reversed towing torque of each lubricating oil was measured twice, and the speed increased from 2000 r/min to 3000 r/min, and then from 3000 r/min to 2000 r/min. The speed interval corresponding to each measurement point was 200 r/

**Table 4. Lubricating oil test standards and instruments.**

| | Measuring standard | instrument |
|---|---|---|
| Density | GB/T 1884–2000 | Anton Paar GmbHDMA 4200M |
| Flash point | GB/T 3536–2008 | Closed flash testerSYD-3536-1 |
| Pour point | GB/T 3535–2006 | Automatic freezing and tipping point testerA1120 |
| Total acid number | GB/T 7304, ASTMD664 | Total acid number titratorAquamax |
| Viscosity-temperature characteristic | GB/T 265–88 | Rotational viscometer brookfield DV-2 pro |

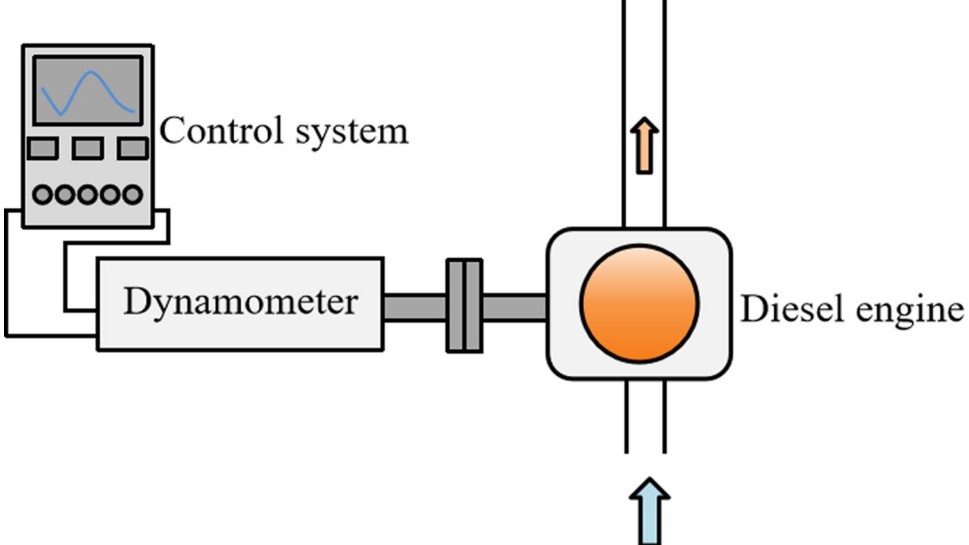

**Fig 2. Schematic diagram of engine reversed towing experiment device.**

min, and the average value of the two measurements was the reversed towing torque. It should be noted that in order to exclude the influence of the viscosity-temperature effect of the lubricating oil on the friction force, the temperature of the lubricating oil was maintained between 80˚C and 85˚C. After the reversed towing experiment of PLO, the engine bench performance experiment of PLO was followed.

## 2.4 Engine bench performance experiment

**2.4.1. Experiment device and instrument.** After the reversed towing experiment of PLO was completed, fuel-metering device and exhaust gas analyzer were added to measure fuel consumption and emissions. Their specific model parameters are shown in Table 6. The schematic diagram of the engine bench performance experiment instruments are shown in Fig 3.

**2.4.2. Experiment scheme.** The characteristics of effective fuel consumption and exhaust temperature change with the engine load lubricated with different lubricating oil at 1000 r/min, 1200 r/min, 1400 r/min, 1600 r/min, 1800 r/min, 2000 r/min and 2200 r/min were compared. Average values were measured 3 times at each working point. The emission performance comparison experiment was carried out in accordance with the standard, Emission Limits and Measurement Methods of Exhaust Pollutants of Diesel

Engines for Non-road Machinery (China III, IV Stage), mainly measuring CO, HC, $NO_X$ and PM emission. After the engine bench experiment of PLO, the engine PM online measurement and collection test of PLO were carried out.

**Table 5. Main instruments and equipment for experiment.**

| Serial number | Name | Model | Manufacturer |
|---|---|---|---|
| 1 | Eddy current dynamometer | CWF75 | Hangzhou Zhongcheng test Equipment Co., LTD |
| 2 | Dynamometer control system | EST2010 | Hangzhou Zhongcheng test Equipment Co., LTD |
| 3 | Lubricating oil, exhaust temperature meter | DT-A | Shanghai Internal Combustion Engine Research Institute |
| 4 | barometer | YM3 | Shanghai Mengde Instrument Co., LTD |
| 5 | hygrothermograph | LX8013 | Guangzhou enjoy electronic Co., LTD |

**Table 6. Main instruments for experiment.**

| Serial number | Name | Model | Manufacturer |
|---|---|---|---|
| 1 | Fuel-metering device | YHW-010 | Hangzhou Zhongcheng test Equipment Co., LTD |
| 2 | Exhaust gas analyzer | HoribaMEXA-7200D | AVL, Austria |

## 2.5 Online measurement and collection experiment of PM

**2.5.1 Experiment device and instrument.** The schematic diagram of PM online measurement and collection device is shown in Fig 4. The American EEPS3090 particle size spectrometer was used for PM online measurement. The sampling flow rate was 50 L/min. The position about 10 cm away from the exhaust pipe was used as the sampling point of the EEPS3090 particle size spectrometer. The pipe connected with the exhaust pipe was a stainless steel pipe with a diameter of 6 mm. The pipe wall was smooth without bending, minimizing resistance to PM. A metal mesh sampling device was used to collect PM samples.

**2.5.2 Experiment scheme.** The effect of different lubricating oil on PM particle size was studied after the reversed towing comparison experiment and engine bench performance comparison experiment. In 2003, China issued the standard "Automobile engine Performance Test Method" GB/T19055-2003, which clearly stipulates that the ratio of lubricating oil and fuel consumption at full load and rated speed shall not exceed 0.3%. In order to accelerate the generation and influence of lubricating oil on PM, the lubricating oil was added to the fuel at the mass ratio of 0.5% to burn together. The change of the PM particle size distribution of lubricating oil was measured online, and the appropriate amount of PM was collected for characterization, so as to study the influence of nano-graphene additives on the particle size of PM. The experimental fuel were two kinds of mixed diesel fuel, respectively, diesel added with 0.5% mass fraction of PLO and diesel added with 0.5% mass fraction of MGL25. The lubricating oil in the oil pan was the same as the lubricating oil added to the diesel. The fuel and lubricating oil in the oil pan used in the experiment are shown in Table 7. Lubricating oil and diesel are more easily miscible. When configuring the mixed fuel, the mixing rod was used to stir fully to achieve the effect of uniform mixing.

The specific experimental process was as follows. After the engine idled for half an hour, the formal experiment began when the lubricating oil temperature reached about 80˚C. When

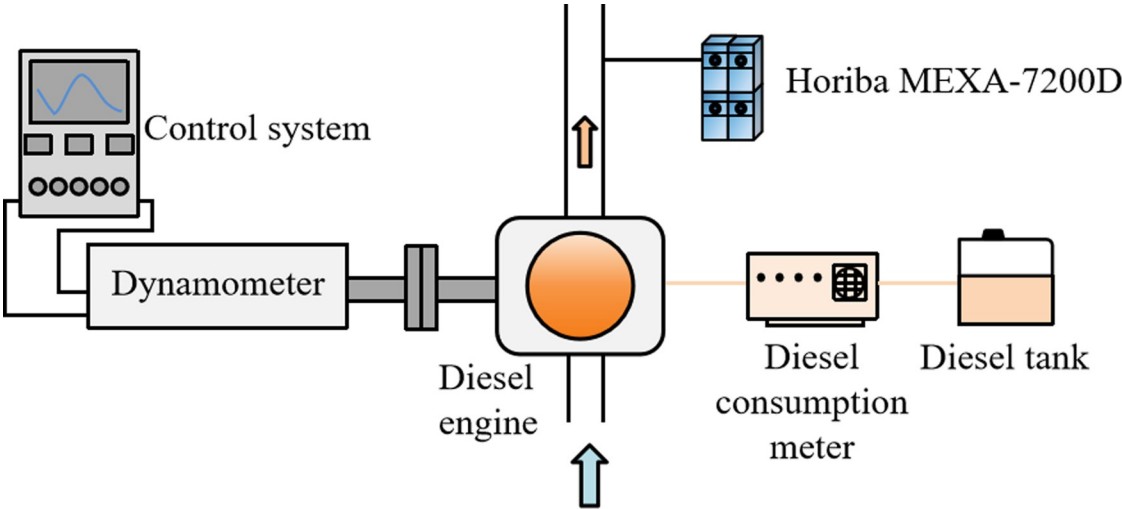

**Fig 3. Schematic diagram of engine bench performance experiment device.**

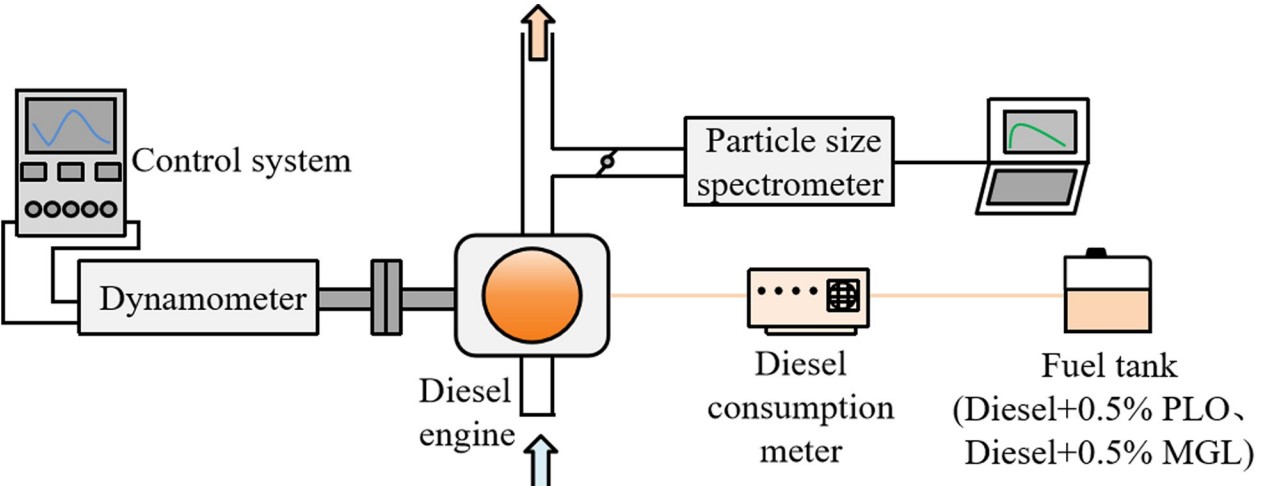

(a) PM online measurement device

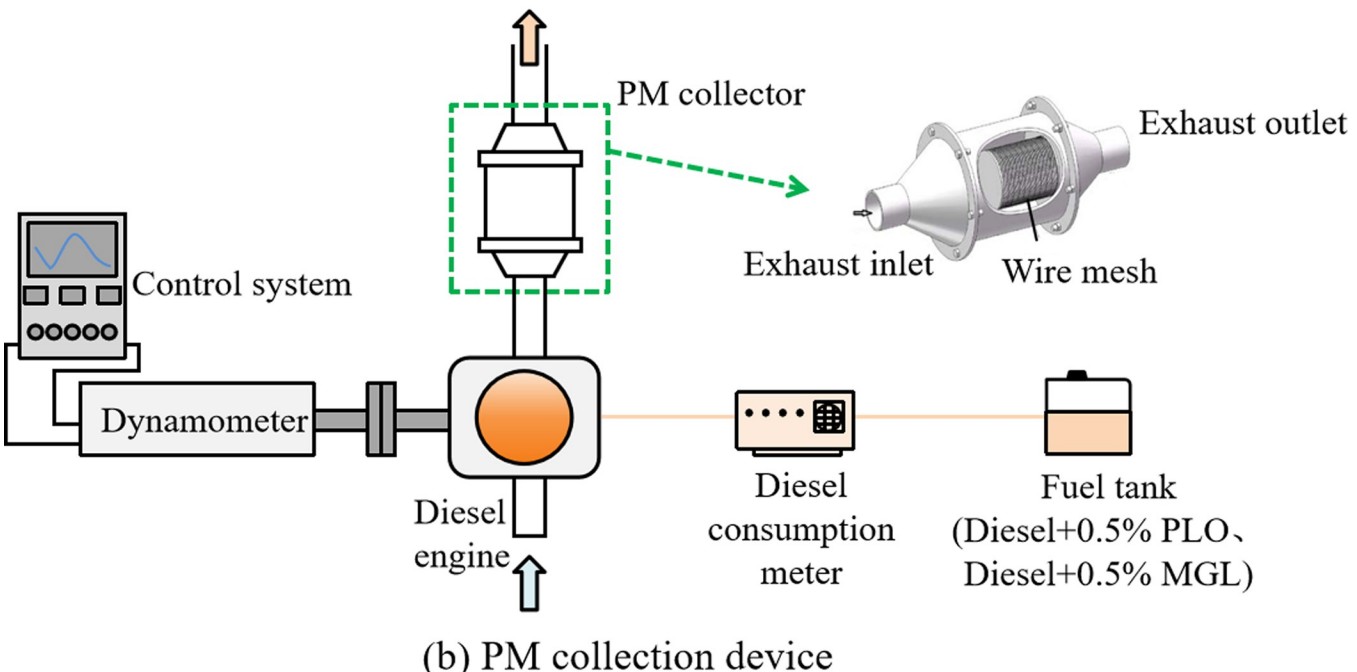

(b) PM collection device

**Fig 4. Schematic diagram of PM experiment device.**

heating the engine, the particle size spectrometer was opened and preheated for about half an hour. The specific experiment operation parameters are shown in Table 8 below. After the engine ran stably for 5 minutes, EEPS was used to sample and measure at each operating

**Table 7. Experimental oil product.**

| Test number | Fuel | Lubricating oil |
| --- | --- | --- |
| 1 | Diesel added with 0.5% mass fraction of PLO | PLO |
| 2 | Diesel added with 0.5% mass fraction of MGL25 | MGL25 |

**Table 8. Experimental conditions.**

| Speed (r/min) | Load |
|---|---|
| 3000 | 10%, 25%, 50%, 75%, 100% |

point. Three measurements were carried out, each with an interval of 30 seconds, and the average value was taken as the final measurement result.

After the completion of the PM online measurement experiment, EEPS was removed, and the self-made sampling device metal mesh was used to collect PM samples. In each experiment, the engine was shut down after stable operation for 90 minutes at the calibration working point. After shutdown, the PM adsorbed in the PM sampling device were scraped off and stored in a clean glassware. JEM-2100(HR) field emission transmission electron microscope was used to capture the morphology of the basic carbon particles of PM. Before testing, the PM samples need to be pre-treated. The treatment method was as follows. A small amount of PM was placed in anhydrous ethanol and subjected to ultrasonic shock for 15 minutes. After standing for 5 minutes, a small amount of the upper layer solution was dropped onto the copper mesh microgrid by pipette. After drying, it was placed on the sample table of high-power transmission electron microscopy for measurement.

When the experiment of one oil product was completed, the fuel in the fuel pipe was released, the lubricating oil in the diesel oil pan was discharged, and fresh experimental fuel and lubricating oil were added. After running the engine, the lubricating oil in the diesel engine oil pan was discharged after running for about half an hour at the speed of 3000 r/min and the torque of 24.2 N·m, when the temperature of water and lubricating oil reached the appropriate value. In this way, the non-experimental lubricating oil in the diesel engine can be cleaned. And re-add the new lubricating oil for the experiment. At the same time, the fuel pipeline was cleaned by the new experimental fuel to ensure that the entire fuel line was filled with new experimental fuel to avoid the interference of the previous fuel on the experimental results.

# 3. Experimental results

## 3.1 Physicochemical properties of nano-graphene lubricating oil

One week later, the dispersion of the nano-graphene lubricating oil before and after modification is shown in Fig 5. It can be obviously seen that after a week, compared with the freshly prepared nano-graphene lubricating oil, the modified nano-graphene lubricating oil has a slightly lighter color and only a little sediment at the bottom of the test tube. While the unmodified nano-graphene lubricating oil has a significantly lighter color and more sediment at the bottom of the test tube. The relative concentration changes of nano-graphene lubricating oil before and after modification are shown in Fig 6. It can be seen that the dispersion stability of the modified nano-graphene lubricating oil is improved, which is consistent with the naked eye. At the same initial concentration, after standing for a week, the relative concentration of unmodified graphene lubricating oil was reduced to 0.262, while the relative concentration of modified graphene lubricating oil was reduced to 0.806. Therefore, the chemical grafting of graphene by oleic acid and stearic acid can significantly improve its dispersion stability in lubricating oil. This is due to the presence of hydroxyl groups on the surface of nanographene, which are esterified with the carboxyl groups of oleic acid and stearic acid. This results in the presence of long alkyl chains on the surface of nanographene, which are more soluble in lubricating oil.

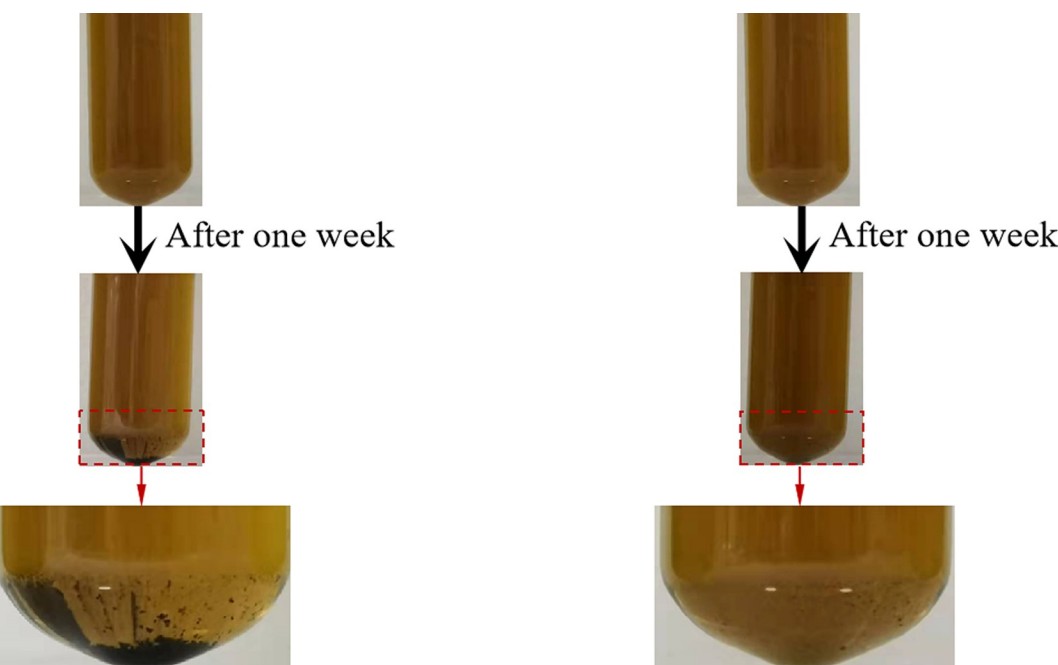

(a) Unmodified nano-graphene lubricating oil  (b) Modified nano-graphene lubricating oil

**Fig 5. Settlement of nano-graphene lubricating oil after one week.**

The kinematic viscosity changes of PLO and MGL25 with temperature are shown in Fig 7. It can be seen that the kinematic viscosity of MGL25 is reduced compared with that of PLO. And with temperatures reaching 60˚C, the decline gradually slows. This is because graphene slips between layers to form self-lubrication, and when the temperature is below 60˚C, the possibility of folding or agglomeration is small when the appropriate amount of graphene is added to the lubricating oil, which is conducive to reducing the internal friction when the lubricating oil fluid flows. When the temperature is higher than 60˚C, some nano-graphene may fold or agglomerate, and the internal friction of lubricating oil decreases slowly.

The measurement results of density, flash point, pour point and total acid value are shown in Table 9. It can be seen from the table that compared with PLO, the density, flash point, pour point and total acid value of nano-graphene lubricating oil have almost no change. This shows that nano-graphene additives have no impact on the safety of lubricating oil. In addition, oleic acid and stearic acid did not affect the total acid value of the nano-graphene lubricating oil by chemical grafting modification of the surface of the nano-graphene additive. The TG curves of different lubricating oils are shown in Fig 8. It can be seen that the weight loss curves of PLO and MGL25 basically coincide. It shows that the addition of nano-graphene has little effect on the oxidation characteristics of lubricating oil. This is because the mass fraction of nano-graphene added is 25 ppm, which is very small and not enough to affect these parameters.

## 3.2 Analysis of reversed towing experiment results

The reversed towing torque of the engine is shown in Fig 9. With the increase of speed, the engine reversed towing torque of different lubricating oil increases. At any speed, the reversed towing torque of nano-graphene lubricating oil is reduced compared with that of PLO. As the speed increases, the decline gradually slows down. This is because the higher the rotational

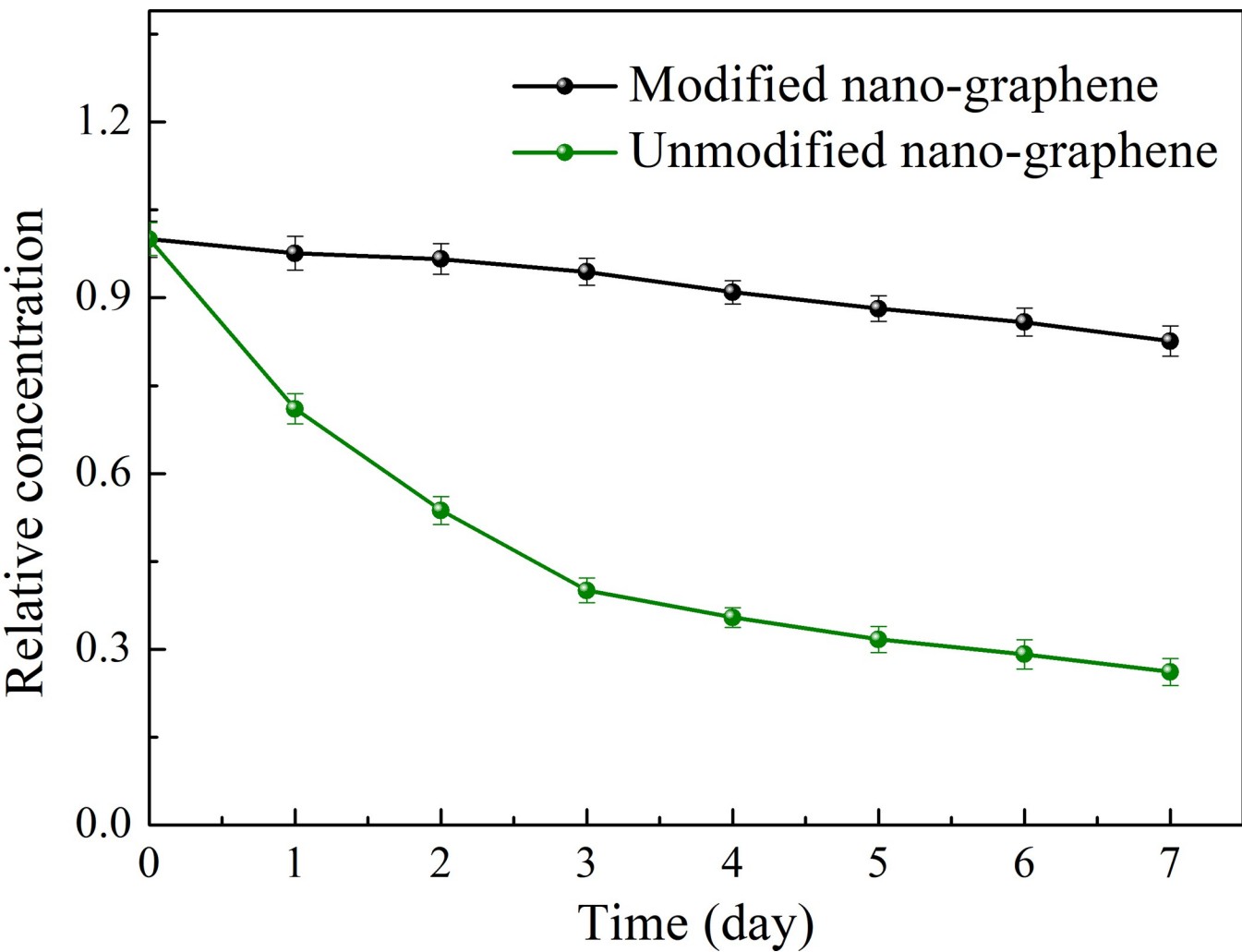

**Fig 6. Relative concentration changes of nano-graphene lubricating oil before and after modification.**

speed, the higher the lubricating oil temperature, and the decreasing trend of lubricating oil viscosity gradually slows down with the increase of temperature. And the fluid friction also shows a similar trend. In the speed range of 2000–3000 r/min, the reversed towing torque reduction rate of MGL25 is 1.82–5.53% compared with that of PLO. This means less friction loss under nano-graphene lubricating oil. The trend of friction reduction in the reversed towing experiment is consistent with the conclusion of the basic tribological experiment [25].

### 3.3 Bench performance experiment

**3.3.1 Comparative experiment of external characteristic.** After nano-graphene was added to the original pure lubricating oil SN/CF 5W-30, the engine worked normally without abnormal phenomenon. The external characteristic curve of the engine lubricated with different lubricating oils is shown in Fig 10. As can be seen from the figure, with the increase of speed, effective fuel consumption first decreases and then increases, and the exhaust temperature increases. Compared with PLO, the effective fuel consumption of the prototype of MGL25 under external characteristic conditions is improved by 2.31–6.01% and the exhaust

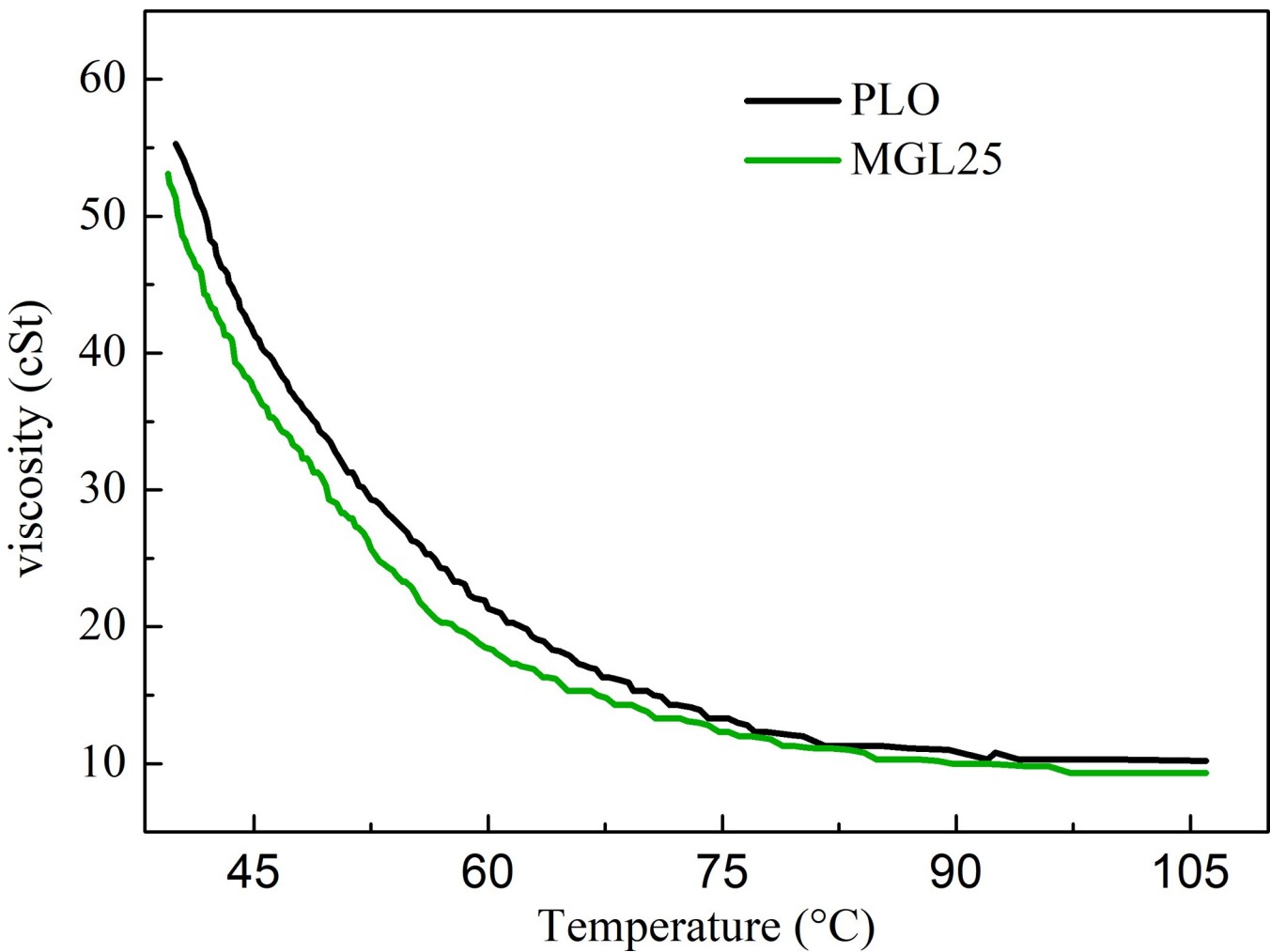

**Fig 7. Variation of kinematic viscosity of different lubricating oils with temperature.**

temperature is reduced by 4.91–15.63%. This shows that the fuel economy is improved after the addition of nano-graphene. The results are consistent with those of the reversed towing experiment. This is due to the reduction of engine friction losses lubricated with MGL25.

**3.3.2 Load characteristic comparison test.** The load characteristic curves of the engine of different lubricating oils are shown in Fig 11 Compared with PLO, the effective fuel consumption of the prototype of MGL25 has decreased. In the load characteristic experiment at 1000, 1200, 1400, 1600, 1800, 2000 and 2200 r/min, the effective fuel consumption is improved by 0.23~2.31% and 2.03~4.39%, 2.25~4.59%, 2.63~ 6.61%, 3.13~5.49%, 3.02 ~ 6.00% and 2.53 ~ 5.26%, respectively; Exhaust temperature decreases to 0.76~6.62%, 5.56~11.72%, 7.61~13.77% and 9.69~12.83%, 9.97~14.49%, 9.16~15.63%, 10.98~15.48%, respectively. This shows that the fuel economy is improved after the addition of nano-graphene. This is due to the fact that

**Table 9. Physicochemical properties parameters of different lubricating oil.**

| Lubricating oil | Density (20°C) (kg/m³) | Flash point (opening) (°C) | Pour point(°C) | Total acid number (in KOH) (mg/g) |
|---|---|---|---|---|
| PLO | 0.893 | 225 | -48 | 1.22 |
| MGL25 | 0.894 | 225 | -48 | 1.23 |

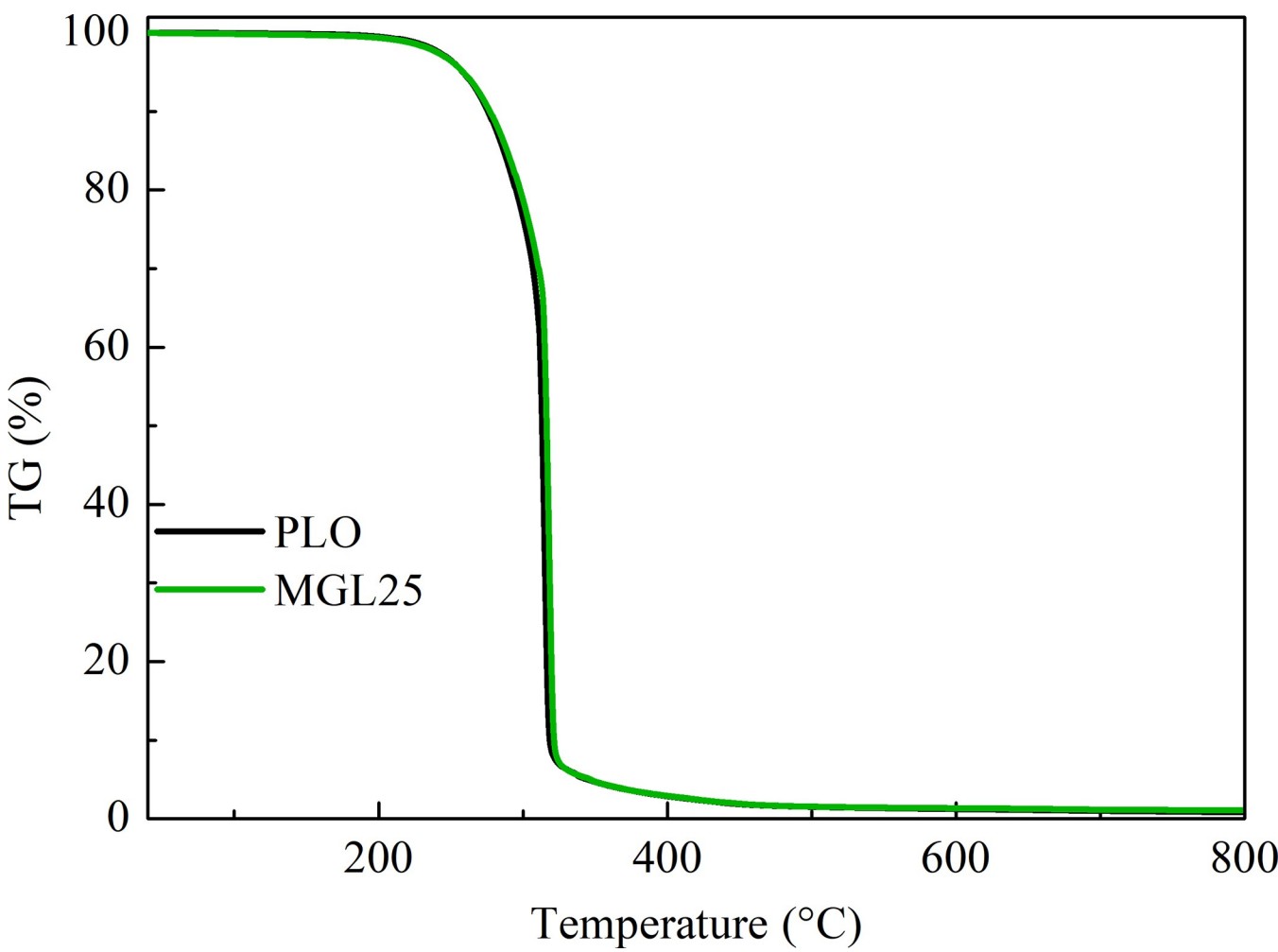

**Fig 8. TG diagram of different lubricating oil.**

nano-graphene reduces friction losses and improves the mechanical efficiency of the engine. The results are consistent with those of the reversed towing experiment. This is due to the reduction of engine friction losses lubricated with nano-graphene lubricating oil.

**3.3.3 Emission performance comparison test.** The engine emission experiment results of different lubricating oils are shown in Table 10. It can be seen from the table that, compared with PLO, CO emission of the prototype of MGL25 is reduced by 11.68%, which is consistent with the research results in the references [29]. HC and $CH_4$ do not change. $NO_X$ and $CO_2$ emissions are reduced by 2.19% and 4.59% respectively, which is similar to the reference results [27]. However, PM emission increases by 8.85%. This shows that the addition of nano-graphene leads to an increase in PM emissions. This is because when the engine is working, a small amount of lubricating oil will inevitably participate in the combustion. Compared with the pure lubricating oil, the viscosity of the nano-graphene lubricating oil is reduced to a certain extent, resulting in a lower mixture concentration of the local combustion chamber, a more complete combustion of the mixture, and a lower CO emission. When the nano-graphene lubricating oil is involved in combustion, the nanoparticles may self-nucleate, and these self-nucleating particles and their agglomeration will be directly adsorbed on the surface of carbon or further agglomerate with carbon particles, resulting in the increase of PM.

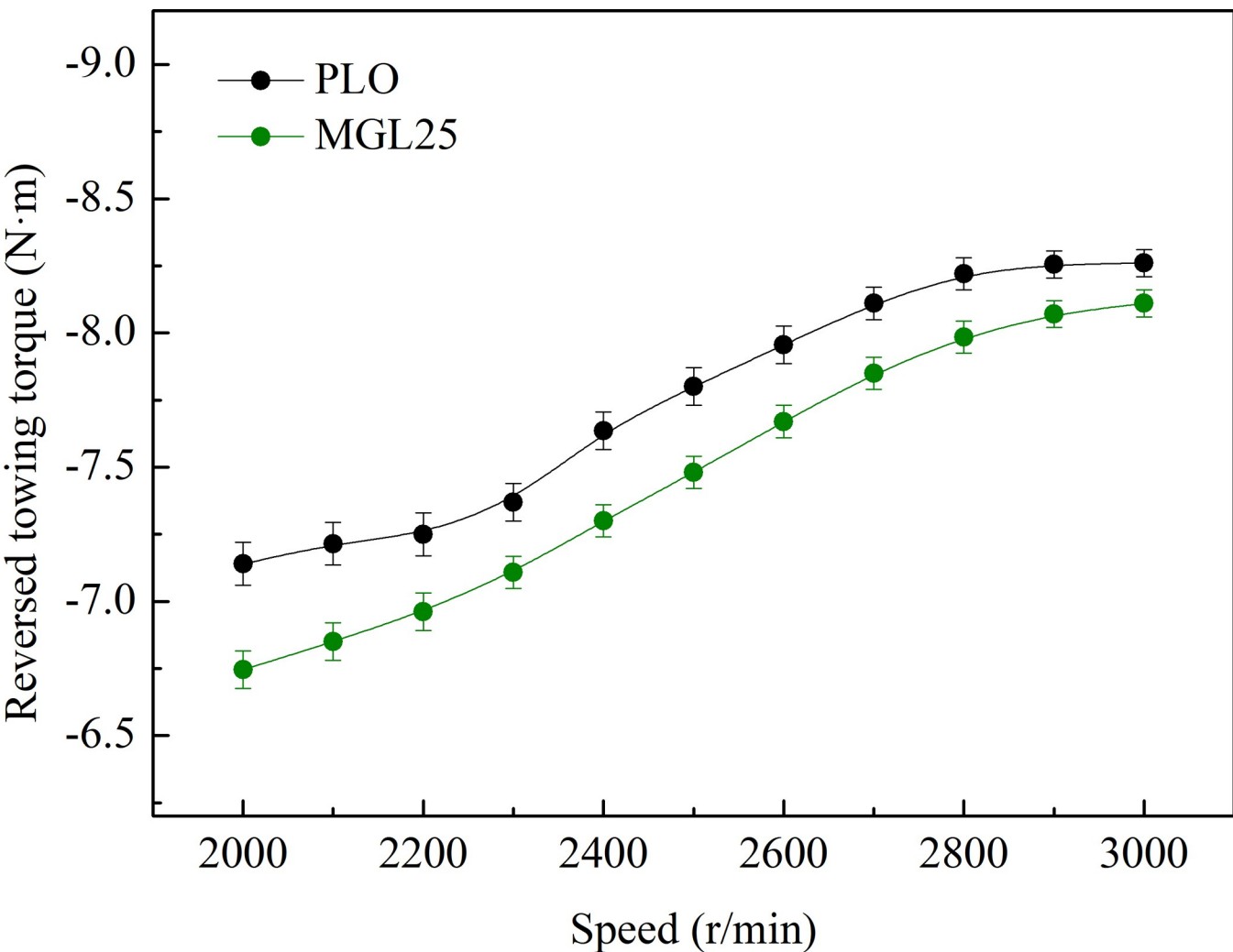

**Fig 9. Reversed towing torque variation diagram.**

### 3.4 Particle size distribution

The particle size distribution results under different torques and different rotational speeds were obtained through the experiment. The experimental results were expressed in the form of $dN/dlogD_p(/cm^3)$, where N is the particle number and $D_p$ is the particle size. The quantity concentration and particle size of PM are related to the running condition of diesel engine, and the logarithmic form is advantageous for comparison. After statistical analysis of the test data, it is found that the particle size between 250–560 nm is almost zero, so the figures only show the particle size distribution between 5.6–250 nm.

**3.4.1. Particle size distribution at rated speed.** Fig 12 respectively shows the changes in the cumulative distribution of PM corresponding to different lubricating oils under different load conditions at the rated speed of 3000 r/min. Particles with a particle size of less than 50 nm are called nuclear particles, which are generally formed by the adsorption of volatile organic matter by the particle core formed by the combination of sulfuric acid vapor and solid carbon particles. Particles in the particle size range of 50 nm to 1000 nm are called aggregated particles, which are composed of nuclear particles that further accumulate into clusters and

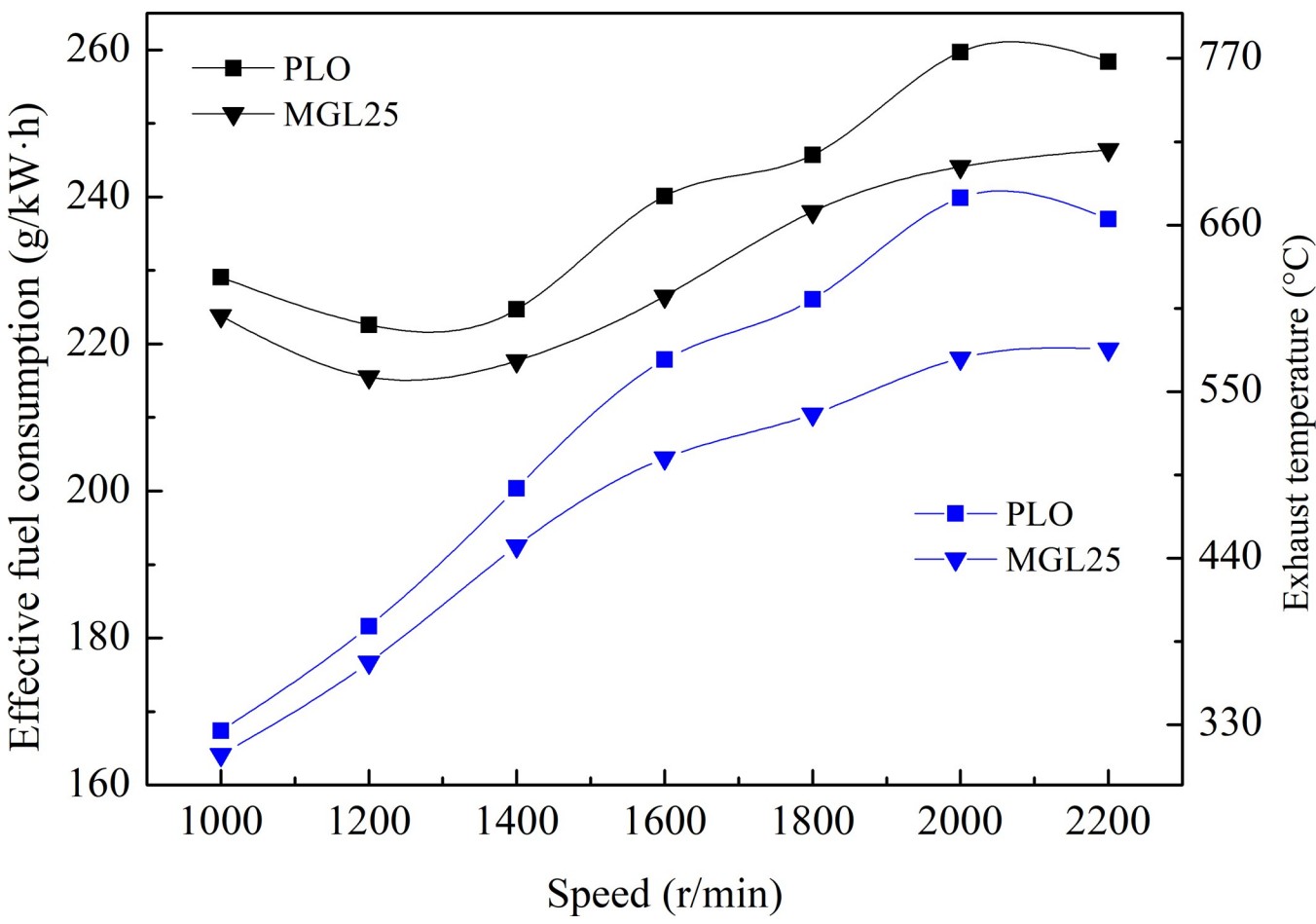

**Fig 10. External characteristic experimental curve.**

condense part of HC and semi-volatile substances such as sulfuric acid. It can be clearly seen from the figure that the concentration of nuclear particles, accumulated particles and total particles of MGL25 is greater than that of PLO, which is consistent with the reference results [30]. And the increase of accumulated particles is larger. And this phenomenon is more obvious at large loads. Studies have shown that particles with a particle size of less than 100 nm can pass through the alveoli and enter the blood, which is very harmful to human health. This means that the disadvantage of nano-graphene lubricating oil is more prominent, and it is necessary to adjust the control strategy of the post-processing system then test and calibrate to reduce the discharge of ultrafine particles into the atmosphere.

Statistically, two important parameters, geometric mean diameter and total quantity concentration, are shown in Fig 13. At the rated speed of diesel engine, the total quantity concentration of particles corresponding to MGL25 increases with the increase of the load. The total quantity concentration of particles corresponding to PLO is larger at full load and small load, while the total quantity concentration of particles corresponding to PLO at intermediate load is smaller. The total quantity concentration of particles of MGL25 is significantly greater than that of PLO. And the larger the load, the more obvious this phenomenon. With the increase of load, the geometric mean diameter of the particles corresponding to the two lubricating oils increases. At different loads, the geometric mean diameter of MGL25 is larger than that of PLO.

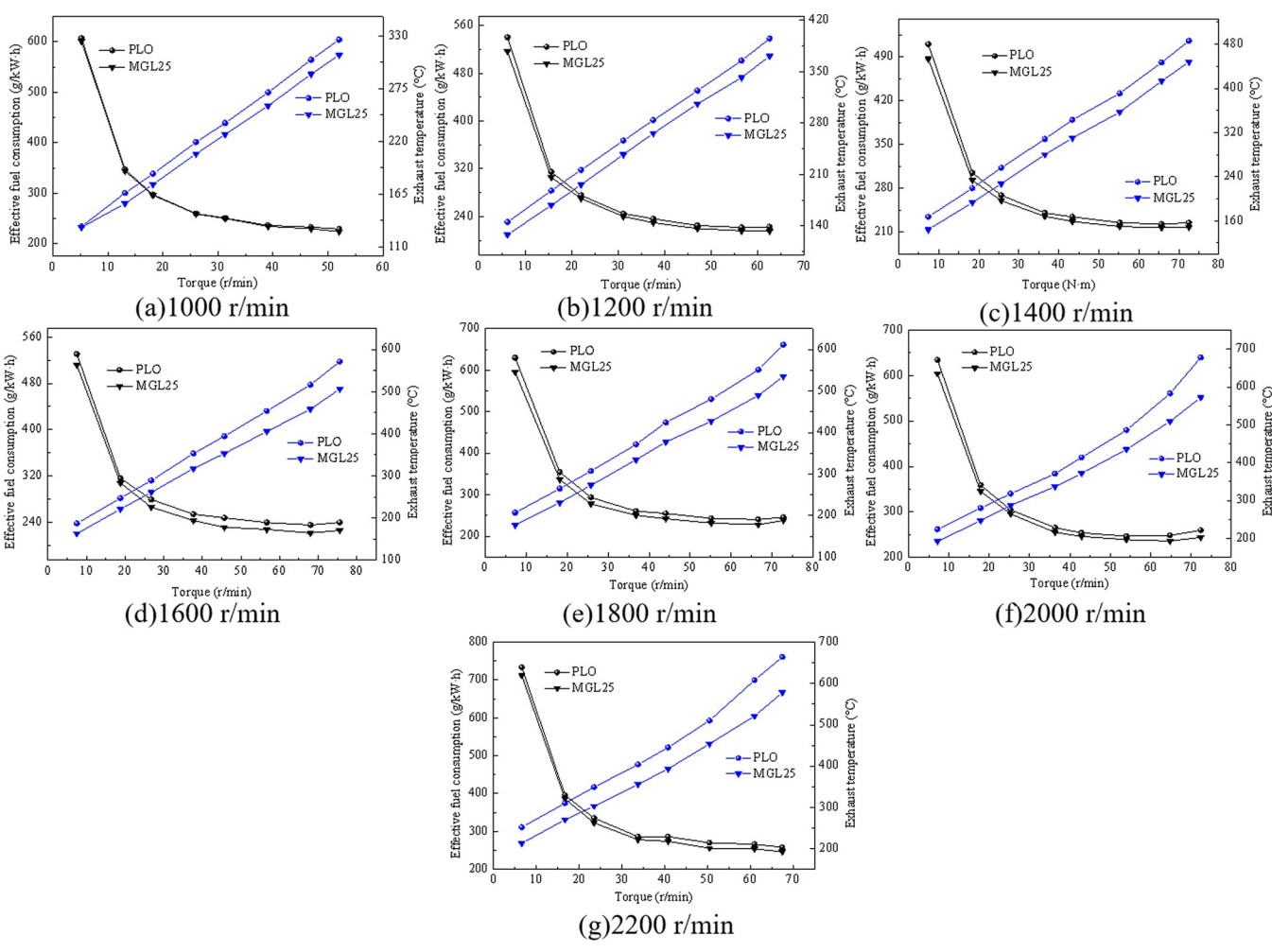

**Fig 11. Load characteristic experiment curve at different speed.**

Under rated conditions, the morphology of PM corresponding to the two lubricating oils at different multiples (46,000, 94000, 190,000 times) is shown in Fig 14. The PM corresponding to the two lubricating oils are composed of dozens to hundreds of basic carbon particles similar to balls, showing irregular shapes such as clusters, chains, branches, etc.. There is not much difference in the intuitive shape, and nano-graphene does not affect the intuitive shape of the particles. It is worth noting that at 94000 times, it is easy to see that the basic carbon particle size of particles of PLO is relatively uniform; However the number of basic carbon particles with smaller particle size of MGL25 increased. The results are consistent with PM online measurements. This is because when the nano-graphene lubricating oil is involved in combustion,

**Table 10. Engine emissions.**

| | CO | THC | $NO_X$ | $CH_4$ | $CO_2$ | NMHC | NMHC+$NO_X$ | PM |
|---|---|---|---|---|---|---|---|---|
| | g/kW·h | g/kW·h | g/kW·h | g/kW·h | g/kW·h | g/kW·h | g/kW·h | g/kW·h |
| National III emission limits | 5.5 | —— | —— | —— | —— | —— | 7.5 | 0.6 |
| PLO | 1.37 | 0.12 | 6.39 | 0.01 | 854.15 | 0.11 | 6.50 | 0.305 |
| MGL25 | 1.21 | 0.12 | 6.25 | 0.01 | 814.99 | 0.12 | 6.37 | 0.332 |

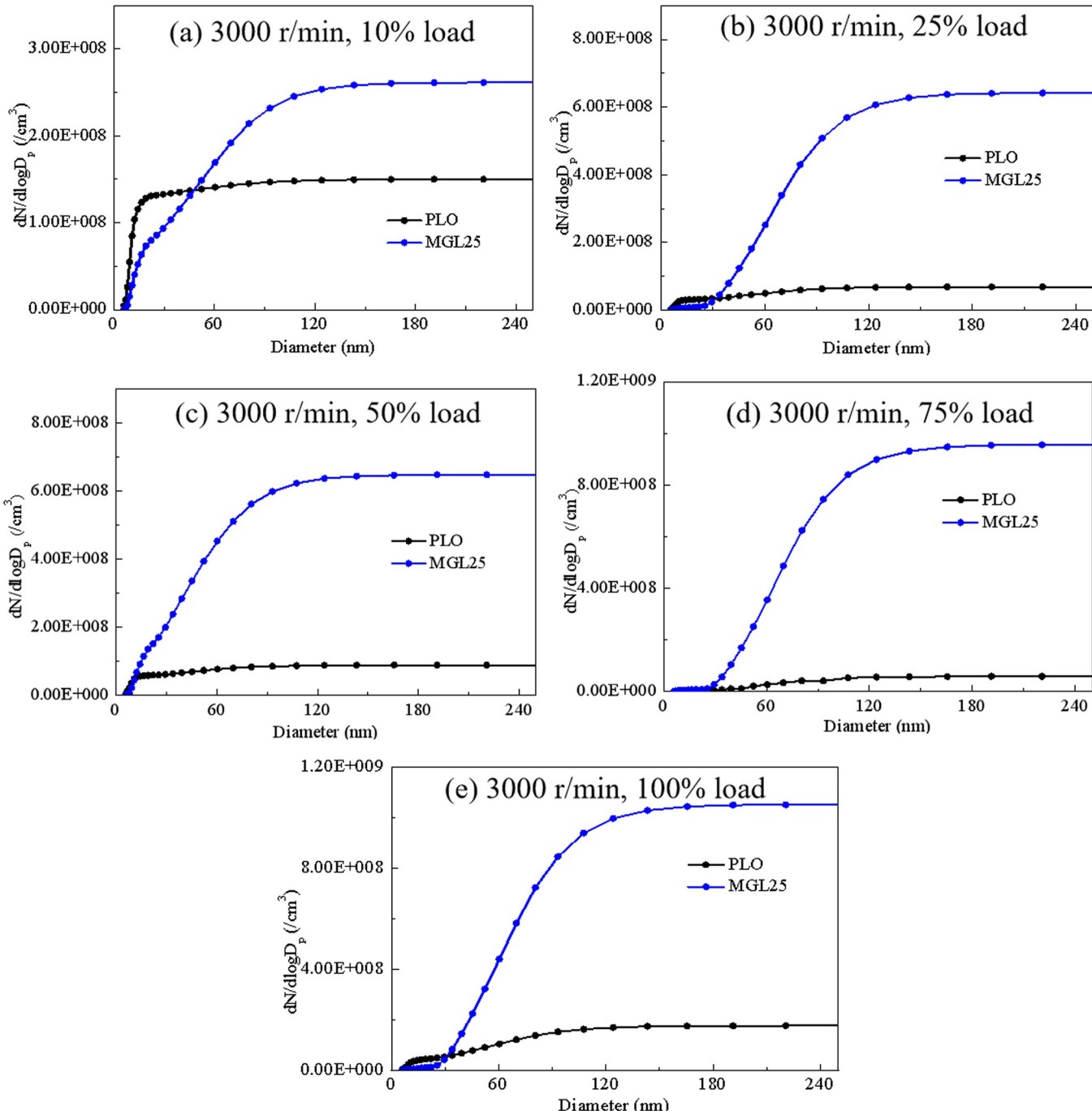

**Fig 12. Cumulative distribution of quantity of PM quantity corresponding to different lubricating oil.**

the nano-particles may self-nucleate, resulting in the risk of increasing nuclear particles. In addition, these self-nucleating particles and their agglomeration will directly adsorb on the carbon surface or further agglomerate with carbon particles to form accumulated particles, resulting in the risk of increasing aggregated particles. And with the increase of load, the temperature increases, resulting in the increase of agglomeration and accumulation of particles.

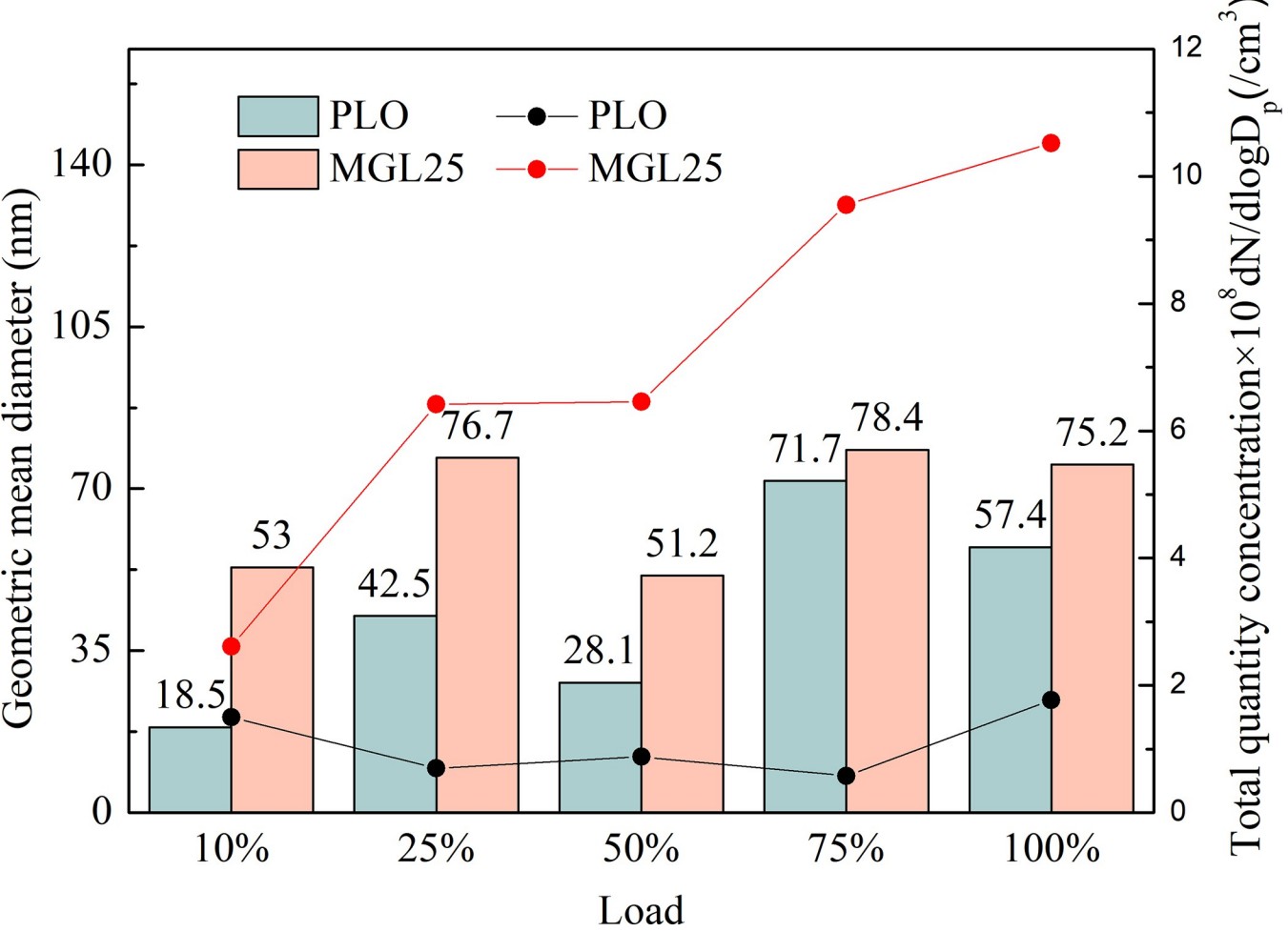

**Fig 13. The geometric mean diameter and total quantity concentration of particulate matter corresponding to different lubricating oils.**

## 4. Conclusions

The real anti-friction effect of nano-graphene lubricating oil and its influence on engine power performance, economic performance and emission performance remain to be investigated. In this paper, nano-graphene powder was chemically modified to prepare nano-graphene lubricating oil, and the effect of nano-graphene on the physicochemical properties of lubricating oil was studied. The effects of nano-graphene on engine power performance, economic performance and emission performance were investigated by engine bench test. The main conclusions are as follows:

1. The dispersion stability of modified nano-graphene lubricating oil is improved.

2. The addition of nano-graphene makes the kinematic viscosity of lubricating oil slightly lower, and has little effect on the density, flash point, pour point and total acid value of lubricating oil.

3. Compared with PLO, the reversed towing torque of nano-graphene lubricating oil is reduced, and the friction loss is reduced.

4. After adding nano-graphene to lubricating oil, the engine fuel economy is improved.

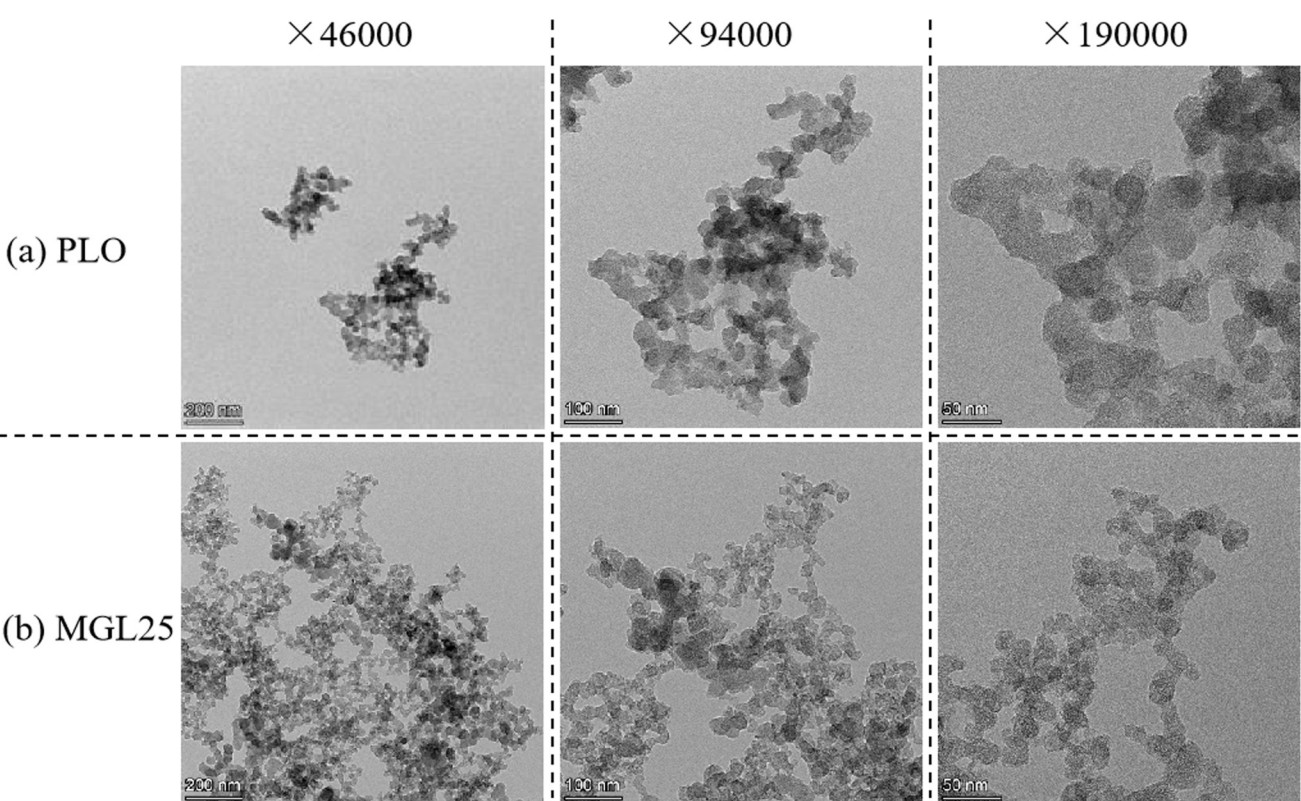

**Fig 14. PM morphologies corresponding to the two lubricating oils at different multiples.**

5. Lubricated with MGL25, the engine emission of HC, $NO_X$, $CH_4$ and $CO_2$ of the engine has little change, but the emission of PM increases by 8.85%.

6. The quantity concentration of nuclear particles and accumulated particles of nano-graphene lubricating oil are significantly higher than that of PLO, and the increase amplitude of the quantity concentration of accumulated particles is more obvious than that of nuclear particles, and the larger the load, the more obvious this phenomenon.

7. The addition of graphene has little effect on the visual morphology of PM. The particle size of basic carbon particles of PM of PLO is relatively uniform, while the number of basic carbon particles with smaller particle size of PM of nano-graphene lubricating oil increases. It confirms that when nano-graphene lubricating oil participates in combustion, the nanoparticles will self-nucleate, increasing the risk of generating more nuclear particles.

In summary, the use of nano-graphene lubricating oil reduces engine friction losses, improves fuel economy, but the number of PM increases, and the nuclear particles increase. Nano-graphene lubricating oil may have adverse effects on post-processing, and further study of its effects on post-processing is needed, as well as re-control strategy adjustment and testing and calibration of the post-processing system. The research has laid a theoretical and scientific basis for the design, development and application of nano-graphene lubricating oil.

## Author Contributions

**Conceptualization:** Xin Kuang.

**Formal analysis:** Hua Bian.

**Investigation:** Xin Kuang.

**Methodology:** Xiping Yang.

**Project administration:** Xin Kuang.

**Supervision:** Xiping Yang.

**Validation:** Xiping Yang.

**Writing – original draft:** Rong Kuang.

**Writing – review & editing:** Nanrong Hu, Shengyong Li.

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
