## [Decision Letter · Decision Letter 0]

26 Apr 2024

PONE-D-24-11357Performance evaluation of nano-graphene lubricating oil with high dispersion and low viscosity used in diesel enginesPLOS ONE

Dear Dr. Xiping Yang,

Thank you for submitting your manuscript to PLOS ONE. After careful consideration, we feel that it has merit but does not fully meet PLOS ONE’s publication criteria as it currently stands. Therefore, we invite you to submit a revised version of the manuscript that addresses the points raised during the review process.

**While preparing the revised manuscript author should take care of the following comment as follows:**

**1. Lacking the novelty of the present work.**

**2. Improve the Introduction section with citing the current published papers.**

**3. Author has not disclose about the thermo-physical properties of nanofluid.**

**4. Also, add the stability evaluation method and its results.**

**5.  Based on comment no. 4 and 5 results incorporate with the engine results. **

We look forward to receiving your revised manuscript.

Kind regards,

Sameer Sheshrao Gajghate, PhD

Academic Editor

PLOS ONE

Journal Requirements:

"This work was supported by Natural Science Research of Jiangsu Higher Education Institutions of China [grant number 23KJB470006；23KJB460009]; Science and Technology Project of Nantong City [grant number JC22022066]；Natural Science Foundation of Jiangsu Province [grant number BK20231227]; High-level Talents Research Start-up Fund supported by Jiangsu Shipping College."

6. We note that Figure 4 in your submission contain copyrighted images. All PLOS content is published under the Creative Commons Attribution License (CC BY 4.0), which means that the manuscript, images, and Supporting Information files will be freely available online, and any third party is permitted to access, download, copy, distribute, and use these materials in any way, even commercially, with proper attribution. For more information, see our copyright guidelines: http://journals.plos.org/plosone/s/licenses-and-copyright.

a. You may seek permission from the original copyright holder of Figure 4 to publish the content specifically under the CC BY 4.0 license. 

Reviewers' comments:

Reviewer's Responses to Questions

**Comments to the Author**

1. Is the manuscript technically sound, and do the data support the conclusions?

Reviewer #1: Yes

Reviewer #2: No

Reviewer #3: Partly

2. Has the statistical analysis been performed appropriately and rigorously? 

Reviewer #1: Yes

Reviewer #2: I Don't Know

Reviewer #3: N/A

3. Have the authors made all data underlying the findings in their manuscript fully available?

Reviewer #1: Yes

Reviewer #2: Yes

Reviewer #3: Yes

4. Is the manuscript presented in an intelligible fashion and written in standard English?

Reviewer #1: Yes

Reviewer #2: Yes

Reviewer #3: No

5. Review Comments to the Author

Reviewer #1: This study presents a well-designed experiment investigating the effects of nano-graphene lubricating oil on engine performance. The research team utilized a controlled environment with varied operating conditions to gather robust data. The findings are particularly encouraging, demonstrating a reduction in friction and improvement in fuel economy with the use of this lubricant. The analysis is further strengthened by the application of advanced techniques for examining particle size distribution. This research paves the way for the development of more efficient and environmentally friendly lubricants.

Reviewer #2: The paper presents valuable experimental data and insights into the use of nano-graphene as a lubricating oil additive in engines. However, further clarification and discussion in certain areas, particularly regarding the implications of engine emissions would enhance the impact and significance of the study. Additionally, suggestions for future research in the direction of tribology could be included to guide further exploration in this field.

Reviewer #3: 1-The research results were not clearly reflected in the abstract, your abstract should clearly state the essence of the problem you are addressing, what you did and what you found and recommend. In addition, please provide final recommendation of your study.

2-1In the introduction, follow the literature review by a little more detailed state of the art analysis.

3-This should clearly show the knowledge gaps identified and link them to your paper goals.

4-Please emphasize both the novelty and the relevance of your paper goals. Correct this.

5-The addition of nano-graphene has little effect on the physicochemical properties

of lubricating oil; explain the reasons.

6-Authors should explicitly specify the novelty of their work.

8-What progress against the most recent state-of-the-art similar studies was made in this study? Mention this in the revised manuscript.

9-The discussion is poor; there is no deep discussion and compared with previous studies.

10-The author should introduce the advantages of graphene. Why its addition will affect the tribological properties and improve the friction?

11-Please proof read again the paper to improve the English standard for the entire manuscript.

12-The authors mention that they achieved good dispersion, but this is not shown. The authors must add microscopic photos showing that they achieved good dispersion and avoided agglomeration.

13-The authors should measure the tribology properties and take the TEM image of the samples after and before the test .

6. PLOS authors have the option to publish the peer review history of their article (what does this mean?). If published, this will include your full peer review and any attached files.

Reviewer #1: **Yes: **Vibhu Sharma

Reviewer #2: No

Reviewer #3: No

---

## [Author Response · Author response to Decision Letter 0]

20 May 2024

Dear Editor,

Many thanks for your email of 26 April 2024, and for your significant efforts to review our manuscript ID PONE-D-24-11357 entitled “Performance evaluation of nano-graphene lubricating oil with high dispersion and low viscosity used in diesel engines”. We would like to thank the reviewers for giving us constructive suggestions which would help us both in English and in depth to improve the quality of the paper. Here we submit a new version, which has been modified according to the reviewers’ suggestions. Efforts were also made to correct the mistakes of the manuscript.

We appreciate your fair consideration of our manuscript and the opportunity to improve our manuscript. We believe that we have modified the manuscript in accordance with all additional issues raised by the reviewers. We hope the manuscript will be acceptable for publication in “Plos One”.

Yours sincerely,

Xiping Yang

The following is a point-to-point response to comments from two reviewers and the academic editor.

Response to reviewer #2:

Reply: Thanks for your significant efforts to review our manuscript and we have carefully studied your comments. According to the review comments you have given,

the manuscript has been revised. The following are responses to your comments.

1.The paper presents valuable experimental data and insights into the use of nano-graphene as a lubricating oil additive in engines. However, further clarification and discussion in certain areas, particularly regarding the implications of engine emissions would enhance the impact and significance of the study. Additionally, suggestions for future research in the direction of tribology could be included to guide further exploration in this field.

We have further clarified and discussed the impact of nano-graphene on engine emissions that will enhance the impact and significance of this study (highlighted in yellow color in line 39-40, 117-119, 123-137, 497-498 of the manuscript). Some suggestions for future tribology research are also put forward (highlighted in yellow color in line 496-499 of the manuscript).

We appreciate for your warm work earnestly, and hope that the correction will meet

with approval.

Once again, thank you very much for your comments and suggestions.

Yours sincerely,

Xiping Yang

Response to reviewer #3:

Reply: Thank you for careful and thorough reading of this manuscript and for the thoughtful comments and constructive suggestions, which help to improve the quality

of this manuscript. We welcome the opportunity to address and clarify the issues raised in the referee report. According to the review comments you have given, the manuscript has been revised. The following are point-to-point responses to your comments.

1. The research results were not clearly reflected in the abstract, your abstract should clearly state the essence of the problem you are addressing, what you did and what you found and recommend. In addition, please provide final recommendation of your study.

We have revised the abstract accordingly (highlighted in blue color in line 48-49, 52-53, 60-63 of the manuscript) .

2. In the introduction, follow the literature review by a little more detailed state of the art analysis.

In the introduction, we have added a more detailed statement of the analysis (highlighted in yellow color in line 85-115, 117-119, 123-137 of the manuscript).

3. This should clearly show the knowledge gaps identified and link them to your paper goals.

We have added the corresponding statement (highlighted in yellow color in line 125, 134-137 of the manuscript).

4. Please emphasize both the novelty and the relevance of your paper goals. Correct this.

We have added the corresponding statement (highlighted in yellow color in line 85-87, 108-109, 117-119, 123-125 of the manuscript).

5. The addition of nano-graphene has little effect on the physicochemical properties

of lubricating oil; explain the reasons.

The addition of nano-graphene has a certain effect on the physicochemical properties of lubricating oil. The addition of nano-graphene reduces the kinematic viscosity of lubricating oil at 25-100°C, which is described in line 331-336 of the manuscript. However, the addition of nano-graphene has little effect on the density, flash point, pour point, total acid value and oxidation characteristics of lubricating oil, which is described in line 339-348 of the manuscript. This is because the mass fraction of nano-graphene added is 25 ppm, which is very small and not enough to affect these parameters. We have revised the relevant statements (highlighted in purple color in line 49-51, 472-474 of the manuscript) and added relevant explanations in the manuscript (highlighted in blue color in line 348-350 of the manuscript).

6. Authors should explicitly specify the novelty of their work.

We have added the corresponding statement (highlighted in yellow color in line 85-87, 117-119, 123-125, 134-137 of the manuscript) to explicitly specify the novelty of their work .

8. What progress against the most recent state-of-the-art similar studies was made in this study? Mention this in the revised manuscript.

We have added the corresponding statement (highlighted in yellow color in line 85-87, 117-119, 123, 125, 134-137 of the manuscript) to explicitly specify the novelty of their work .

9. The discussion is poor; there is no deep discussion and compared with previous studies.

We have added in-depth discussions and comparisons (highlighted in blue color in line 317-323, 348-350, 388-389, 395-397, 412-418, 421-426 of the manuscript) to explicitly specify the novelty of their work .

10. The author should introduce the advantages of graphene. Why its addition will affect the tribological properties and improve the friction?

We have added relevant statements to introduce the advantages of graphene (highlighted in blue color in line 80-82 of the manuscript).

11. Please proof read again the paper to improve the English standard for the entire manuscript.

We have proofread the paper to improve the English standard for the entire manuscript.

12. The authors mention that they achieved good dispersion, but this is not shown. The authors must add microscopic photos showing that they achieved good dispersion and avoided agglomeration.

The dispersion of graphene in lubricating oil after one week was presented respectively before and after modification in line 327 of the manuscript. In order to quantify the dispersion effect of graphene in lubricating oil before and after modification, we made up the test to measure the absorbance change of graphene in lubricating oil before and after modification in a week. (highlighted in blue color in line 184-188, 317-323 of the manuscript). The absorbance change of graphene in lubricating oil can reflect the dispersion problem better than the microscopic photos.

13. The authors should measure the tribology properties and take the TEM image of the samples after and before the test .

In the early stage, our research group has carried out tribological performance testing, TEM image acquisition and related analysis of nano-graphene lubricating oil, and the relevant results have been published in a journal, which is described in the introduction (highlighted in yellow color in line 107-115 of the manuscript).

We appreciate for your warm work earnestly, and hope that the correction will meet with approval.

Once again, thank you very much for your comments and suggestions.

Yours sincerely,

Xiping Yang

Response to the academic editor:

Reply: Thanks for your significant efforts to review our manuscript and we have carefully studied your comments. According to the review comments you have given,

the manuscript has been revised. The following are responses to your comments.

1.Though introduction section clearly highlights the current study's novelty. however, a justification for using graphene as an additive over other nano particles has not been provided.1.

We have added the relevant statement introduction section (highlighted in yellow color in line 80-82 of the manuscript).

2.In Line 95, what is “0#” represents?

0# refers to the diesel label, which is divided according to the freezing point of diesel. 0# diesel is corresponding to the loss of diesel fluidity at no higher than 0°C.

3.In Line 100, Author claims that “The nano-graphene was chemically modified by oleic acid and stearic acid”. Whereas nano-graphene should not be reacted with acids. 

There are oxygen-containing groups, carboxyl groups and hydroxyl groups on the surface of graphene or on the surface of graphene oxide during the preparation of graphene. The oxygen-containing groups on the surface of graphene can undergo ring-opening reactions with stearic acid under the action of catalysts. The hydroxyl group on the surface of graphene and the carboxyl group of oleic acid can be esterified under the condition of catalyst. Oil-soluble graphene can be obtained by grafting long-chain alkanes onto the surface of graphene.

4.In Line 102, Author claims that “2g stearic acid and 3 g oleic acid were then added into them” It is not cleared about the purpose of adding the acids. Again, why only these two particular types of the acids are added?

The purpose of adding stearic acid and oleic acid is to graft oleic acid and oleic acid onto the surface of nano-graphene to achieve oil solubility. This is stated in the first line 157-158 of the manuscript, with relevant references. 

5.Comparing Table 2 and 3: Density of Diesel is 836 kg/m3 and density of PLO is 0.893 kg/ m3. If so, then how the flash point of PLO is significantly higher than diesel. 

We are very sorry that we made a mistake. The density of diesel should be 0.836 kg/m3, not 836 kg/m3. We have corrected the mistake in line 173 of the manuscript (highlighted in blue color).

6.Line 208: “The experimental fuel were two kinds of mixed diesel fuel, respectively, diesel added with 0.5% mass fraction of PLO and diesel added with 0.5% mass fraction of MGL25.” Elaborate the process if possible.

This process was described in line 263-265, 271-274, 306 of the manuscript.

7.Line 215: “After the engine idled for half an hour, the formal experiment began when the lubricating oil temperature reached about 80°C.” already given in line Line:157.

Here are two experiments. They are the engine reversed towing experiment and the on-line measurement experiment of particulate matter. Both experiments require the oil to reach 80°C before starting formal experiments.

8.Table 10 represents the engine emissions where as the unit given in g/KW-h. How that can be possible?

According to the objects, contents and detection methods tested in different standards, the emission units are not the same. This study was in accordance with the "Non-road machinery diesel engine exhaust pollutant emission limits and measurement methods (China III, IV)" standard, the emission unit is g/kWh. It refers to the quality of pollutants emitted by the engine per unit time per unit power, that is a specific emission.

9.Moreover, From the table 10 it is found NOx emission is significantly lower for MGL25, whereas Fig. 10 represents fuel consumption for MGL25 is lower throughout the speed variation, that means higher efficiency of the engine. Why this controversy occurs?

MGL corresponds to a 2.19% reduction in NOX emissions compared to pure lubricating oil, a result consistent with lower exhaust temperature. The addition of nanographene has little effect on NOX. And this result is consistent with the reference [27]. The MGL corresponding fuel consumption is reduced by 0.23% to 6.61%, which is due to the nano-graphene reducing friction losses and improving the mechanical efficiency of the engine. We have added relevant descriptions in line 388-389, 395-397 of the manuscript (highlighted in blue color).

[27]Ali MKA, Xianjun H, Abdelkareem MA, Gulzar M, Elsheikh A. Novel approach of the graphene nanolubricant for energy saving via anti-friction/wear in automobile engines. Tribology International. 2018;124:209-29.

10.The quality of Fig.11 should be improved.

We have already improved the quality of Fig. 11.

11.In conclusion claimed about the good antifriction effect of nano-graphene lubricant over generally used lubricant. Whereas no comparison of tribological behaviours between these two lubricants have not found.

Our research group has done basic tribological experiments in the early stage, compared the tribological properties of pure lubricating oil and nano-graphene lubricating oil, and published a paper. We have added relevant descriptions in 107-115 lines of the manuscript (highlighted in yellow color). In this study, on the basis of the results of the basic tribological experiment, we studied the friction power consumption under the lubrication of the two lubricating oils by the reversed towing test. The results show that the antifriction effect of nano-graphene lubricating oil is better than that of pure lubricating oil, which is consistent with the results of basic tribological experimental.

We appreciate for your warm work earnestly, and hope that the correction will meet

with approval.

Once again, thank you very much for your comments and suggestions.

Yours sincerely,

Xiping Yang

---

## [Decision Letter · Decision Letter 1]

24 Jun 2024

PONE-D-24-11357R1Performance evaluation of nano-graphene lubricating oil with high dispersion and low viscosity used in diesel enginesPLOS ONE

Dear Dr. Yang,

Thank you for submitting your manuscript to PLOS ONE. After careful consideration, we feel that it has merit but does not fully meet PLOS ONE’s publication criteria as it currently stands. Therefore, we invite you to submit a revised version of the manuscript that addresses the points raised during the review process.<ul>The uncertainty Analysis section is missing throughout the manuscript.Nomenclature section is missing. 

Add the physio-chemical properties of the lubricants

We look forward to receiving your revised manuscript.

Kind regards,

Sameer Sheshrao Gajghate, PhD

Academic Editor

PLOS ONE

Reviewers' comments:

Reviewer's Responses to Questions

**Comments to the Author**

1. If the authors have adequately addressed your comments raised in a previous round of review and you feel that this manuscript is now acceptable for publication, you may indicate that here to bypass the “Comments to the Author” section, enter your conflict of interest statement in the “Confidential to Editor” section, and submit your "Accept" recommendation.

Reviewer #2: All comments have been addressed

Reviewer #4: All comments have been addressed

2. Is the manuscript technically sound, and do the data support the conclusions?

Reviewer #2: Yes

Reviewer #4: No

3. Has the statistical analysis been performed appropriately and rigorously? 

Reviewer #2: Yes

Reviewer #4: No

4. Have the authors made all data underlying the findings in their manuscript fully available?

Reviewer #2: Yes

Reviewer #4: Yes

5. Is the manuscript presented in an intelligible fashion and written in standard English?

Reviewer #2: Yes

Reviewer #4: No

6. Review Comments to the Author

Reviewer #2: (No Response)

Reviewer #4: The language should be improved

The authors should clearly correlate The experimental results with basic theory

A reduction of 1 to 5 percent was found in the reverse towing torque. I doubt The accuracy of experiment.

The authors must include uncertainty analysis

Most important property for lubricants. Viscosity index was omitted in physico chemical properties

7. PLOS authors have the option to publish the peer review history of their article (what does this mean?). If published, this will include your full peer review and any attached files.

Reviewer #2: No

Reviewer #4: **Yes: **Dr. Vadapalli Srinivas

---

## [Author Response · Author response to Decision Letter 1]

27 Jun 2024

Dear Editor,

Many thanks for your email of 25 June 2024, and for your significant efforts to review our manuscript ID PONE-D-24-11357R1 entitled “Performance evaluation of nano-graphene lubricating oil with high dispersion and low viscosity used in diesel engines”. We would like to thank the reviewers for giving us constructive suggestions which would help us both in English and in depth to improve the quality of the paper. Here we submit a new version, which has been modified according to the reviewers’ suggestions. Efforts were also made to correct the mistakes of the manuscript.

We appreciate your fair consideration of our manuscript and the opportunity to improve our manuscript. We believe that we have modified the manuscript in accordance with all additional issues raised by the reviewers. We hope the manuscript will be acceptable for publication in “Plos One”.

Yours sincerely,

Xiping Yang

The following is a point-to-point response to comments from two reviewers and the academic editor.

Response to reviewer #4:

Reply: Thanks for your significant efforts to review our manuscript and we have carefully studied your comments. According to the review comments you have given,

the manuscript has been revised. The following are responses to your comments.

1.The language should be improved.

We have revised the language (highlighted in purple color in line 55-56, 70-71, 80, 84-87, 94-98, 125, 129, 135, 160, 164, 179, 191-192, 196, 198, 205, 209-210, 214, 216, 224, 228, 258-259, 263-264, 276, 279, 294, 323, 363, 365-366, 371, 374, 389, 404, 473 of the manuscript).

2. The authors should clearly correlate The experimental results with basic theory

We have added the corresponding description and analysis (highlighted in green color in line 325-329, 337-339, 342-344, 367-371, 388-390, 403-405, 414-422 of the manuscript).

3. A reduction of 1 to 5 percent was found in the reverse towing torque. I doubt The accuracy of experiment.

The authors must include uncertainty analysis

We have added the specific description of the measurement of the reversed towing torque in the experiment scheme. (highlighted in yellow color in line 239 of the manuscript). And an error bar was added to the reversed towing torque variation diagram. (highlighted in yellow color in line 377 of the manuscript).

4. Most important property for lubricants. Viscosity index was omitted in physico chemical properties

In the last revised manuscript, the relevant detection and analysis of lubricating oil viscosity index have been described. (highlighted in blue color in line 183-186, 335-336, 346 of the manuscript).

We appreciate for your warm work earnestly, and hope that the correction will meet

with approval.

Once again, thank you very much for your comments and suggestions.

Yours sincerely,

Xiping Yang

Response to the academic editor:

Reply: Thank you for careful and thorough reading of this manuscript and for the thoughtful comments and constructive suggestions, which help to improve the quality

of this manuscript. We welcome the opportunity to address and clarify the issues raised in the referee report. According to the review comments you have given, the manuscript has been revised. The following are point-to-point responses to your comments.

1.The uncertainty Analysis section is missing throughout the manuscript.

We have added the specific description (highlighted in yellow color in line 239 of the manuscript). And error bars were added to diagrams (highlighted in yellow color in line 333, 377 of the manuscript).

2.Nomenclature section is missing

The nomenclature section for PLO and MGL25 is described in line 167-168 (highlighted in blue color of the manuscript)

3.Add the physio-chemical properties of the lubricants.

The physio-chemical properties of the lubricants, including viscosity index, were described and analyzed in the original manuscript (highlighted in blue color in line 183-193, 335-336, 346 of the manuscript).

We appreciate for your warm work earnestly, and hope that the correction will meet

with approval.

Once again, thank you very much for your comments and suggestions.

Yours sincerely,

Xiping Yang

---

## [Decision Letter · Decision Letter 2]

4 Jul 2024

Performance evaluation of nano-graphene lubricating oil with high dispersion and low viscosity used in diesel engines

PONE-D-24-11357R2

Dear Dr. Xiping Yang,

We’re pleased to inform you that your manuscript has been judged scientifically suitable for publication and will be formally accepted for publication once it meets all outstanding technical requirements.

Kind regards,

Sameer Sheshrao Gajghate, PhD

Academic Editor

PLOS ONE

Additional Editor Comments (optional):

Reviewers' comments:

Reviewer's Responses to Questions

**Comments to the Author**

1. If the authors have adequately addressed your comments raised in a previous round of review and you feel that this manuscript is now acceptable for publication, you may indicate that here to bypass the “Comments to the Author” section, enter your conflict of interest statement in the “Confidential to Editor” section, and submit your "Accept" recommendation.

Reviewer #2: All comments have been addressed

Reviewer #4: All comments have been addressed

2. Is the manuscript technically sound, and do the data support the conclusions?

Reviewer #2: Yes

Reviewer #4: Yes

3. Has the statistical analysis been performed appropriately and rigorously? 

Reviewer #2: Yes

Reviewer #4: Yes

4. Have the authors made all data underlying the findings in their manuscript fully available?

Reviewer #2: Yes

Reviewer #4: Yes

5. Is the manuscript presented in an intelligible fashion and written in standard English?

Reviewer #2: Yes

Reviewer #4: Yes

6. Review Comments to the Author

Reviewer #2: (No Response)

Reviewer #4: the authors have adequately addressed the comments raised in a previous round of review. the manuscript technically sound, and the data support the conclusions

7. PLOS authors have the option to publish the peer review history of their article (what does this mean?). If published, this will include your full peer review and any attached files.

Reviewer #2: No

Reviewer #4: **Yes: **v srinivas

---

## [Editor Report · Acceptance letter]

7 Aug 2024

PONE-D-24-11357R2 

PLOS ONE

Dear Dr. Yang, 

I'm pleased to inform you that your manuscript has been deemed suitable for publication in PLOS ONE. Congratulations! Your manuscript is now being handed over to our production team.

Kind regards, 

on behalf of

Dr. Sameer Sheshrao Gajghate 

Academic Editor

PLOS ONE